# An Efficient Streaming Algorithm
# for the Submodular Cover Problem

**Ashkan Norouzi-Fard** *
ashkan.norouzifard@epfl.ch

**Abbas Bazzi** *
abbas.bazzi@epfl.ch

**Marwa El Halabi** †
marwa.elhalabi@epfl.ch

**Ilija Bogunovic** †
ilija.bogunovic@epfl.ch

**Ya-Ping Hsieh** †
ya-ping.hsieh@epfl.ch

**Volkan Cevher** †
volkan.cevher@epfl.ch

## Abstract

We initiate the study of the classical Submodular Cover (SC) problem in the data streaming model which we refer to as the Streaming Submodular Cover (SSC). We show that any single pass streaming algorithm using sublinear memory in the size of the stream will fail to provide any non-trivial approximation guarantees for SSC. Hence, we consider a relaxed version of SSC, where we only seek to find a *partial* cover. We design the first *Efficient bicriteria Submodular Cover Streaming* (ESC-Streaming) algorithm for this problem, and provide theoretical guarantees for its performance supported by numerical evidence. Our algorithm finds solutions that are competitive with the near-optimal offline greedy algorithm despite requiring only a *single* pass over the data stream. In our numerical experiments, we evaluate the performance of ESC-Streaming on active set selection and large-scale graph cover problems.

## 1 Introduction

We consider the *Streaming Submodular Cover (SSC)* problem, where we seek to find the smallest subset that achieves a certain utility, as measured by a monotone submodular function. The data is assumed to arrive in an arbitrary order and the goal is to minimize the number of passes over the whole dataset while using a memory that is as small as possible.

The motivation behind studying SSC is that many real-world applications can be modeled as cover problems, where we need to select a small subset of data points such that they maximize a particular utility criterion. Often, the quality criterion can be captured by a utility function that satisfies submodularity [27, 16, 15], an intuitive notion of diminishing returns.

Despite the fact that the standard *Submodular Cover* (SC) problem is extensively studied and very well-understood, all the proposed algorithms in the literature heavily rely on having access to whole ground set during their execution. However, in many real-world applications, this assumption does not hold. For instance, when the dataset is being generated on the fly or is too large to fit in memory, having access to the whole ground set may not be feasible. Similarly, depending on the application, we may have some restrictions on how we can access the data. Namely, it could be that random access to the data is simply not possible, or we might be restricted to only accessing a small fraction of it. In all such scenarios, the optimization needs to be done on the fly.

The SC problem is first considered by Wolsey [28], who shows that a simple greedy algorithm yields a logarithmic factor approximation. This algorithm performs well in practice and usually returns

solutions that are near-optimal. Moreover, improving on its theoretical approximation guarantee is not possible under some natural complexity theoretic assumptions [12, 10]. However, such an offline greedy approach is impractical for the SSC, since it requires an infeasible number of passes over the stream.

## 1.1 Our Contribution

In this work, we rigorously show that achieving any *non-trivial* approximation of SSC with a single pass over the data stream, while using a *reasonable* amount of memory, is *not* possible. More generally, we establish an uncontidional lower bound on the trade-off between the memory and approximation ratio for any $p$-pass streaming algorithm solving the SSC problem.

Hence, we consider instead a relaxed version of SSC, where we only seek to achieve a fraction $(1 - \epsilon)$ of the specified utility. We develop the first *Efficient bicriteria Submodular Cover Streaming* (ESC-Streaming) algorithm. ESC-Streaming is simple, easy to implement, and memory as well time efficient. It returns solutions that are competitive with the near-optimal offline greedy algorithm. It requires only a single pass over the data in arbitrary order, and provides for any $\epsilon > 0$, a $2/\epsilon$-approximation to the optimal solution, while achieving a $(1 - \epsilon)$ fraction of the specified utility. In our experiments, we test the performance of ESC-Streaming on active set selection in materials science and graph cover problems. In the latter, we consider a graph dataset that consists of more than $787$ million nodes and $47.6$ billion edges.

## 1.2 Related work

Submodular optimization has attracted a lot of interest in machine learning, data mining, and theoretical computer science. Faced with streaming and massive data, the traditional (offline) greedy approaches fail. One popular approach to deal with the challenge of the data deluge is to adopt streaming or distributed perspective. Several submodular optimization problems have been studied so far under these two settings [25, 11, 9, 2, 20, 8, 18, 7, 14, 1]. In the streaming setting, the goal is to find nearly optimal solutions, with a minimal number of passes over the data stream, memory requirement, and computational cost (measured in terms of oracle queries).

A related streaming problem to SSC was investigated by Badanidiyuru et al. [2], where the authors studied the streaming Submodular Maximization (SM) problem *subject to a cardinality constraint*. In their setting, given a budget $k$, the goal is to pick at most $k$ elements that achieve the largest possible utility. Whereas for the SC problem, given a utility $Q$, the goal is to find the minimum number of elements that can achieve it. In the offline setting of cadinality constrained SM, the greedy algorithm returns a solution that is $(1 - 1/e)$ away from the optimal value [21], which is known to be the best solution that one can obtain efficiently [22]. In the streaming setting, Badanidiyuru et al. [2] designed an elegant single pass $(1/2 - \epsilon)$-approximation algorithm that requires only $O((k \log k)/\epsilon)$ memory. More general constraints for SM have also been studied in the streaming setting, e.g., in [8].

Moreover, the *Streaming Set Cover* problem, which is a special case of the SSC problem is extensively studied [25, 11, 9, 7, 14, 1]. In this special case, the elements in the data stream are $m$ subsets of a universe $\mathcal{X}$ of size $n$, and the goal is to find the minimum number of sets $k^*$ that can cover all the elements in the universe $\mathcal{X}$. The study of the *Streaming Set Cover* problem is mainly focused on the *semi-streaming* model, where the memory is restricted to $\widetilde{O}(n)$[3]. This regime is first investigated by Saha and Getoor [25], who designed a $O(\log n)$-pass, $O(\log n)$-approximation algorithm that uses $\widetilde{O}(n)$ space. Emek and Rosén [11] show that if one restricts the streaming algorithm to perform only one pass over the data stream, then the best possible approximation guarantee is $O(\sqrt{n})$. This lower bound holds even for randomized algorithms. They also designed a deterministic greedy algorithm that matches this approximation guarantee. By relaxing the single pass constraint, Chakrabarti and Wirth [7] designed a $p$-pass semi-streaming $(p + 1)n^{1/(p+1)}$-approximation algorithm, and proved that this is essentially tight up to a factor of $(p + 1)^3$.

***Partial* streaming submodular optimization.** The *Streaming Set Cover* has also been studied from a bicriteria perspective, where one settles for solutions that only cover a $(1-\epsilon)$-fraction of the universe. Building on the work of [11], the authors in [7] designed a semi-streaming $p$-pass streaming algorithm

that achieves a $(1-\epsilon, \delta(n, \epsilon))$-approximation, where $\delta(n, \epsilon) = \min\{8p\epsilon^{1/p}, (8p+1)n^{1/(p+1)}\}$. They also provided a lower bound that matches their approximation ratio up to a factor of $\Theta(p^3)$.

**Distributed submodular optimization.** Mirzasoleiman et al. [20] consider the SC problem in the distributed setting, where they design an efficient algorithm whose solution is close to that of the offline greedy algorithm. Moreover, they study the trade-off between the communication cost and the number of rounds to obtain such a solution.

To the best of our knowledge, no other works have studied the general SSC problem. We propose the first efficient algorithm ESC-Streaming that approximately solves this problem with tight guarantees.

## 2  Problem Statement

**Preliminaries.** We assume that we are given a utility function $f : 2^V \mapsto \mathbb{R}^+$ that measures the quality of a given subset $S \subseteq V$, where $V = \{e_1, \cdots, e_m\}$ is the ground set. The marginal gain associated with any given element $e \in V$ with respect to some set $S \subseteq V$, is defined as follows

$$\Delta_f(e|S) := \Delta(e|S) = f(S \cup \{e\}) - f(S).$$

In this work, we focus on *normalized*, *monotone*, *submodular* utility functions $f$, where $f$ is referred to be:

1. **submodular** if for all $S, T$, such that $S \subseteq T$, and for all $e \in V \setminus T$, $\Delta(e|S) \geq \Delta(e|T)$.
2. **monotone** if for all $S, T$ such that $S \subseteq T \subseteq V$, we have $f(S) \leq f(T)$.
3. **normalized** if $f(\emptyset) = 0$.

In the standard *Submodular Cover* (SC) problem, the goal is to find the smallest subset $S \subseteq V$ that satisfies a certain utility $Q$, i.e.,

$$\min_{S \subseteq V} |S| \text{ s.t. } f(S) \geq Q. \tag{SC}$$

**Hardness results.** The SC problem is known to be NP-Hard. A simple greedy strategy [28] that in each round selects the element with the highest marginal gain until $Q$ is reached, returns a solution of size at most $H(\max_e f(\{e\}))k^*$, where $k^*$ is the size of the optimal solution set $S^*$.[4] Moreover, Feige [12] proved that this is the best possible approximation guarantee unless $NP \subseteq DTIME\left(n^{O(\log\log n)}\right)$. This was recently improved to an NP-hardness result by Dinur and Steurer [10].

**Streaming Submodular Cover (SSC).** In the streaming setting, the main challenge is to solve the SC problem while maintaining a small memory and without performing a large number of passes over the data stream. We use $m$ to denote the size of the data stream. Our first result states that any single pass streaming algorithm with an approximation ratio better than $m/2$, must use at least $\Omega(m)$ memory. Hence, for large datasets, if we restrict ourselves to a single pass streaming algorithm with sublinear memory $o(m)$, we cannot obtain *any* non-trivial approximation of the SSC problem (cf., Theorem 2 in Section 4). To obtain non-trivial and feasible guarantees, we need to relax the coverage constraint in SC. Thus, we instead solve the *Streaming Bicriteria Submodular Cover* (SBSC) defined as follows:

**Definition 1.** *Given $\epsilon \in (0, 1)$ and $\delta \geq 1$, an algorithm is said to be a $(1 - \epsilon, \delta)$-bicriteria approximation algorithm for the SBSC problem if for any Submodular Cover instance with utility $Q$ and optimal set size $k^*$, the algorithm returns a solution $S$ such that*

$$f(S) \geq (1 - \epsilon)Q \quad and \quad |S| \leq \delta k^*. \tag{1}$$

## 3  An efficient streaming submodular cover algorithm

**ESC-Streaming algorithm.** The first phase of our algorithm is described in Algorithm 1. The algorithm receives as input a parameter $M$ representing the size of the allowed memory. The

discussion of the role of this parameter is postponed to Section 4. The algorithm keeps $t + 1 = \log(M/2) + 1$ *representative* sets. Each representative set $S_j$ ($j = 0, .., t$) has size at most $2^j$, and has a corresponding threshold value $Q/2^j$. Once a new element $e$ arrives in the stream, it is added to all the representative sets that are not fully populated, and for which the element's marginal gain is above the corresponding threshold, i.e., $\Delta(e|S_j) \geq \frac{Q}{2^j}$. This phase of the algorithm requires only one pass over the data stream. The running time of the first phase of the algorithm is $O(\log(M))$ for every element of the stream, since the per-element computational cost is $O(\log(M))$ oracle calls.

In the second phase (i.e., Algorithm 2), given a feasible $\tilde{\epsilon}$, the algorithm finds the smallest set $S_i$ among the stored sets, such that $f(S_i) \geq (1 - \tilde{\epsilon})Q$. For any query, the running time of the second phase is $O(\log\log(M))$. Note that after one-pass over the stream, we have no limitation on the number of the queries that we can answer, i.e., we do not need another pass over the stream. Moreover, this phase does not require any oracle calls, and its total memory usage is at most $M$.

---

**Algorithm 1** ESC-Streaming Algorithm - Picking representative set

---
$t = \log(M/2)$
 1: $S_0 = S_1 = ... = S_t = \emptyset$
 2: **for** $i = 1, \cdots, m$ **do**
 3:     Let $e$ be the next element in the stream
 4:     **for** $j = 0, \cdots, t$ **do**
 5:         **if** $\Delta(e|S_j) \geq \frac{Q}{2^j}$ and $|S_j| \leq 2^j$ **then**
 6:             $S_j \leftarrow S_j \cup e$
 7:         **end if**
 8:     **end for**
 9: **end for**

---

**Algorithm 2** ESC-Streaming Algorithm - Responding to the queries

---
Given value $\tilde{\epsilon}$, perform the following steps
 1: Run a binary search on the $S_0, ..., S_t$
 2: Return the smallest set $S_i$ such that $f(S_i) \geq (1 - \tilde{\epsilon})Q$
 3: If no such set exists, Return "Assumption Violated"

---

In the following section, we analyze ESC-Streaming and prove that it is an $(1 - \tilde{\epsilon}, 2/\tilde{\epsilon})$-bicriteria approximation algorithm for SSC. Formally we prove the following:

**Theorem 1.** *For any given instance of SSC problem, and any values $M, \tilde{\epsilon}$, such that $k^*/\tilde{\epsilon} \leq M$, where $k^*$ is size optimal solution to SSC, ESC-Streaming algorithm returns a $(1 - \tilde{\epsilon}, 2/\tilde{\epsilon})$-approximation solution.*

## 4  Theoretical Bounds

**Lower Bound.**  We start by establishing a lower bound on the tradeoff between the memory requirement and the approximation ratio of any $p$-pass streaming algorithm solving the SSC problem.

**Theorem 2.** *For any number of passes $p$ and any stream size $m$, a $p$-pass streaming algorithm that, with probability at least $2/3$, approximates the submodular cover problem to a factor smaller than $\frac{m^{\frac{1}{p}}}{p+1}$, must use a memory of size at least $\Omega\left(\frac{m^{\frac{1}{p}}}{p(p+1)^2}\right)$.*

The proof of this theorem can be found in the supplementary material.

Note that for $p = 1$, Theorem 2 states that any one-pass streaming algorithm with an approximation ratio better than $m/2$ requires at least $\Omega(m)$ memory. Hence, for large datasets, Theorem 2 rules out *any* approximation of the streaming submodular cover problem, if we restrict ourselves to a one-pass streaming algorithm with sublinear memory $o(m)$. This result motivates the study of the *Streaming Bicriterion Submodular Cover* (SBSC) problem as in Definition 1.

**Main result and discussion.**  Refining the analysis of the greedy algorithm for SC [28], where we stop once we have achieved a utility of $(1 - \epsilon)Q$, yields that the number of elements that we

pick is at most $k^* \ln(1/\epsilon)$. This yields a *tight* $(1 - \epsilon, \ln(1/\epsilon))$-bicriteria approximation algorithm for the *Bicriteria Submodular Cover* (BSC) problem. One can turn this bicriteria algorithm into an $(1 - \epsilon, \ln(1/\epsilon))$-bicriteria algorithm for SBSC, at the cost of doing $k^* \ln(1/\epsilon)$ passes over the data stream which may be infeasible for some applications. Moreover, this requires $mk^* \ln\left(\frac{1}{\epsilon}\right)$ oracle calls, which may be infeasible for large datasets.

To circumvent these issues, it is natural to parametrize our algorithm by a user defined memory budget $M$ that the streaming algorithm is allowed to use. Assuming, for some $0 < \epsilon \le e^{-1}$, that the $(1 - \epsilon, \ln(1/\epsilon))$-bicriteria solution given by the *offline* greedy algorithm's fits in a memory of $M/2$ for the BSC variant of the problem, then our algorithm (ESC-Streaming) is guaranteed to return a $(1 - 1/\ln(1/\epsilon), 2\ln(1/\epsilon))$-bicriteria solution for the SBSC problem, while using at most $M$ memory. Hence in only one pass over the data stream, ESC-Streaming returns solutions guaranteed to cover, for small values of $\epsilon$, almost the same fraction of the utility as the greedy solution, loosing only a factor of two in the worst case solution size. Moreover, the number of oracle calls needed by ESC-Streaming is only $m \log M$, which for $M = 2k^* \ln(1/\epsilon)$ is bounded by

$$ m \log M = \underbrace{m \log(2k^* \ln(1/\epsilon))}_{\text{oracle calls by ESC-Streaming algorithm}} \ll \underbrace{mk^* \ln\left(\frac{1}{\epsilon}\right)}_{\text{oracle calls by greedy}}, $$

which is more than a factor $k^*/\log(k^*)$ smaller than the greedy algorithm. This enables ESC-Streaming algorithm to perform much faster than the offline greedy algorithm. Another feature of ESC-Streaming is that it performs a single pass over the data stream, and after this unique pass, we are able to query a $(1 - 1/\ln(1/\epsilon'), 2\ln(1/\epsilon'))$-bicriteria solution for any $\epsilon \le \epsilon' \le e^{-1}$, without any additonal oracle calls. Whenever the above inequality does not hold, ESC-Streaming returns "Assumption Violated". More precisely, we state the following Theorem whose proof can be found in the supplementary material.

**Theorem 3.** *For any given instance of SSC problem, and any values $M, \epsilon$, such that $2k^* \ln 1/\epsilon \le M$, where $k^*$ is the optimal solution size, ESC-Streaming algorithm returns a $(1 - 1/(\ln 1/\epsilon), 2\ln 1/\epsilon)$-approximation solution.*

**Remarks.** Note that in Algorithm 1, we can replace the constant 2 by another choice of the constant $1 < \alpha \le 2$. The representative sets are changed accordingly to $\alpha^j$, and $t = \log_\alpha(M/\alpha)$. Varying $\alpha$ provides a trade-off between memory and solution size guarantee. More precisely, for any $1 < \alpha \le 2$, ESC-Streaming achieves a $(1 - \frac{1}{\ln 1/\epsilon}, \alpha \ln 1/\epsilon)$-approximation guarantee, for instances of SSC where $\alpha k^* \ln 1/\epsilon \le M$. However, the improvement in the size approximation guarantee, comes at the cost of increased memory usage $\frac{M-1}{\alpha-1}$, and increased number of oracle calls $m(\log_\alpha(M/\alpha) + 1)$.

Notice that in the statement of Theorem 3, the approximation guarantee of ESC-Streaming is given with respect to a memory only large enough to fit the offline greedy algorithm's solution. However, if we allow our memory $M$ to be as large as $k^*/\epsilon$, then Theorem 1 follows immediately for $\tilde{\epsilon} = 1/\ln(1/\epsilon)$.

# 5 Example Applications

Many real-world problems, such as data summarization [27], image segmentation [16], influence maximization in social networks [15], can be formulated as a submodular cover problem and can benefit from the streaming setting. In this section, we discuss two such concrete applications.

## 5.1 Active set selection

To scale kernel methods (such as kernel ridge regression, Gaussian processes, etc.) to large data sets, we often rely on active set selection methods [23]. For example, a significant problem with Gaussian process prediction is that it scales as $\mathcal{O}(n^3)$. Storing the kernel matrix $K$ and solving the associated linear system is prohibitive when $n$ is large. One way to overcome this is to select a small subset of data while maintaining a certain diversity. A popular approach for active set selection is Informative Vector Machine (IVM) [26], where the goal is to select a set $S$ that maximizes the utility function

$$ f(S) = \frac{1}{2} \log \det(I + \sigma^{-2} K_{S,S}),. \tag{2} $$

Here, $K_{S,S}$ is the submatrix of $K$, corresponding to rows/columns indexed by $S$, and $\sigma > 0$ is a regularization parameter. This utility function is monotone submodular, as shown in [17].

## 5.2 Graph set cover

In a lot of applications, e.g., influence maximization in social networks [15], community detection in graphs [13], etc., we are interested in selecting a small subset of vertices from a massive graph that "cover" in some sense a large fraction of the graph.

In particular, in section 6, we consider two fundamental set cover problems: Dominating Set and Vertex Cover problems. Given a graph $G(V, E)$ with vertex set $V$ and edge set $E$, let $\rho(S)$ denote the neighbours of the vertices in $S$ in the graph, and $\delta(S)$ the edges in the graph connect to a vertex in $S$. The dominating set is the problem of selecting the smallest set that covers the vertex set $V$, i.e., the corresponding utility is $f(S) = |\rho(S) \cup S|$. The vertex cover is the problem of selecting the smallest set that covers the edge set $E$, i.e., the corresponding utility is $f(S) = |\delta(S)|$. Both utilities are monotone submodular functions.

# 6  Experimental Results

We address the following questions in our experiments:

1. How does ESC-Streaming perform in comparison to the offline greedy algorithm, in terms of solution size and speed?

2. How does $\alpha$ influence the trade-off between solution size and speed ?

3. How does ESC-Streaming scale to massive data sets?

We evaluate the performance of ESC-Streaming on real-world data sets with two applications: active set selection and graph set cover problems, described in section 5. For active set selection, we choose a dataset having a size that permits the comparison with the offline greedy algorithm. For graph cover, we run ESC-Streaming on a large graph of 787 million nodes and 47.6 billion edges.

We measure the computational cost in terms of the number of oracle calls, which is independent of the concrete implementation and platform.

## 6.1  Active Set Selection for Quantum Mechanics

In quantum chemistry, computing certain properties, such as atomization energy of molecules, can be computationally challenging [24]. In this setting, it is of interest to choose a small and diverse training set, from which one can predict the atomization energy (e.g., by using kernel ridge regression) of other molecules.

In this setting, we apply ESC-Streaming on the log-det function defined in Section 5.1 where we use the Gaussian kernel $K_{ij} = \exp(-\frac{\|x_i - x_j\|_2^2}{2h^2})$, and we set the hyperparameters as in [24]: $\sigma = 1, h = 724$. The dataset consists of 7k small organic molecules, each represented by a 276 dimensional vector. We set $M = 2^{15}$ and vary $Q$ from $\frac{f(V)}{2}$ to $\frac{3f(V)}{4}$, and $\alpha$ from 1.1 to 2.

We compare against offline greedy, and its accelerated version with lazy updates (Lazy Greedy)[19]. For all algorithms, we provide a vector of different values of $\tilde{\epsilon}$ as input, and terminate once the utility $(1 - \tilde{\epsilon})Q$, corresponding to the smallest $\tilde{\epsilon}$, is achieved. Below we report the performance for the smallest and largest tested $\tilde{\epsilon} = 0.01$ and $\tilde{\epsilon} = 0.5$, respectively.

In Figure 6.1, we show the performance of ESC-Streaming with respect to the offline greedy and lazy greedy, in terms of size of solutions picked and number of oracle calls made. The computational costs of all algorithms are normalized to those of offline greedy.

It can be seen that standard ESC-Streaming, with $\alpha = 2$, always chooses a set at most twice (largest ratio is 2.1089) as large as offline greedy, using at most 3.15% and 25.5% of the number of oracle calls made, respectively, by offline greedy and lazy greedy. As expected, varying the parameter $\alpha$ leads to smaller solutions at the cost of more oracle calls: $\alpha = 1.1$ leads to solutions roughly of the same size as the solutions found by the offline greedy. Note also that choosing larger values $\alpha$ leads to jumps in the solution sets sizes (c.f., 6.1). In particular, varying the required utility $Q$, even by a

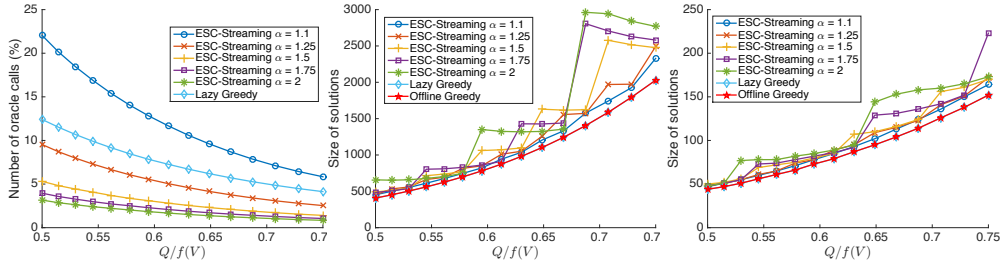

Figure 6.1: Active set selection of molecules: (Left) Percentage of oracle calls made relative to offline greedy, (Middle) Size of selected sets for $\epsilon = 0.01$, (Right) Size of selected sets for $\epsilon = 0.5$.

small amount, may not be possible to achieve by the current solution's size $(\alpha^j)$ and would require moving to a set larger by at least an $\alpha$ factor $(\alpha^{j+1})$.

Finally, we remark that even for this small dataset, offline greedy, for largest tested $Q$, required $1.2 \times 10^7$ oracle calls, and took almost 2 days to run on the same machine.

### 6.2 Cover problems on Massive graphs

To assess the scalability of ESC-Streaming, we apply it to the "uk-2014" graph, a large snapshot of the .uk domain taken at the end of 2014 [5, 4, 3]. It consists of $787,801,471$ nodes and $47,614,527,250$ edges. This graph is sparse, with average degree $60.440$, and hence requires large cover solutions. Storing this dataset (i.e., the adjacency list of the graph) on the hard-drive requires more than 190GB of memory.

We solve both the Dominating Set and Vertex Cover problems, whose utility functions are defined in Section 5. For the Dominating Set problem, we set $M = 520$ MB, $\alpha = 2$ and $Q = 0.7|V|$. We run the first phase of ESC-Streaming (c.f., Algorithm 1), then query for different values of $\tilde{\epsilon}$ between 0 to 1, using Algorithm 2. Similarly, for the Vertex Cover problem, we set $M = 320$ MB, $\alpha = 2$ and $Q = 0.8|E|$. Figure 6.2 shows the performance of ESC-Streaming on both the dominating set and vertex cover problems, in terms of utility achieved, i.e., number vertices/edges covered, for all the feasible $\tilde{\epsilon}$ values, with respect to the size of the subset of vertices picked.

As a baseline, we compare against a random selection procedure, that picks a random permutation of the vertices and then select any vertex with a non-zero marginal, until it reaches the same partial cover achieved by ESC-Streaming. Note that the offline greedy, even with lazy evaluations, is not applicable here since it does not terminate in a reasonable time, so we omit it from the comparison. Similarly, we do not compare against the Emek–Rosén's algorithm [11], due to its large memory requirement of $n \log m$, which in this case is roughly 20 times bigger than the memory used by ESC-Streaming.

We do significantly better than a random selection, especially on the Vertex Cover problem, which for sparse graphs is more challenging than the Dominating Set problem.

Since running the greedy algorithm on "uk-2014" graph goes beyond our computing infrastructure, we include another instance of the Dominating set cover problem on a smaller graph "Friendster", an online gaming network [29], to compare with offline greedy algorithm. This graph has $65.6$ million nodes, and $1.8$ billion edges. The memory required by ESC-Streaming is less than 30MB for $\alpha = 2$. We let offline greedy run for 2 days, and gathered data for 2000 greedy iterations. Figure 6.2 (Right) shows that our performance almost matches the greedy solutions we managed to compute.

## 7 Conclusion

In this paper, we consider the SC problem in the streaming setting, where we select the least number of elements that can achieve a certain utility, measured by a submodular function. We prove that there cannot exist any single pass streaming algorithm that can achieve a non-trivial approximation

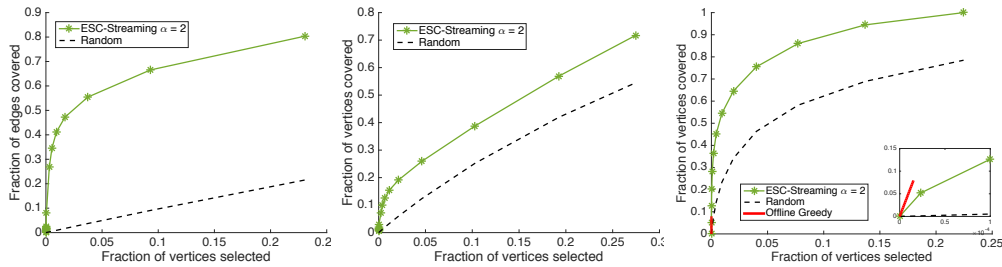

Figure 6.2: (Left) Vertex cover on "uk-2014" (Middle) Dominating set on "uk-2014" (Right) Dominating set on "Friendster"

of SSC, using sublinear memory, if the utility have to be met exactly. Consequently, we develop an efficient approximation algorithm, ESC-Streaming, which finds solution sets, slightly larger than the optimal solution, that partially cover the desired utility. We rigorously analyzed the approximation guarantees of ESC-Streaming, and compared these guarantees against the offline greedy algorithm. We demonstrate the performance of ESC-Streaming on real-world problems. We believe that our algorithm is an important step towards solving streaming and large scale submodular cover problems, which lie at the heart of many modern machine learning applications.

### Acknowledgments

We would like to thank Michael Kapralov and Ola Svensson for useful discussions. This work was supported in part by the European Commission under ERC Future Proof, SNF 200021-146750, SNF CRSII2-147633, NCCR Marvel, and ERC Starting Grant 335288-OptApprox.

## Footnotes

*Theory of Computation Laboratory 2 (THL2), EPFL. These authors contributed equally to this work.

†Laboratory for Information and Inference Systems (LIONS), EPFL

[3]The $\widetilde{O}$ notation is used to hide poly-log factors, i.e., $\widetilde{O}(n) := O(n \operatorname{poly}\{\log n, \log m\})$

[4]Here, $H(x)$ is the $x$-th harmonic number and is bounded by $H(x) \leq 1 + \ln x$.

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
