[Supplementary Material]

## A  Proof of Theorem 3

For convenience, we state Theorem 3 again here, and present its proof.

**Theorem 4.** *For any given instance of SSC problem, and any values $M, \epsilon$, such that $2k^* \ln 1/\epsilon \leq M$, where $k^*$ is the optimal solution size, ESC-Streaming algorithm returns a $(1 - 1/(\ln 1/\epsilon), 2 \ln 1/\epsilon)$-approximation solution.*

*Proof.* Choose $j$ such that $\frac{2^j}{2 \ln 1/\epsilon} \leq k^* < \frac{2^{j+1}}{2 \ln 1/\epsilon}$. We first show that the number of elements in $S_j$ is at most $2k^* \ln 1/\epsilon$ and $f(S_j) \geq (1 - \frac{1}{\ln 1/\epsilon})Q$. We know that $|S_j| \leq 2^j$, and hence

$$|S_j| \leq 2^j \leq 2k^* \ln 1/\epsilon$$

If $|S_j| = 2^j$ then we get $f(S_j) \geq Q$. Now assume that $|S_j| < 2^j$ and let $S^*$ be the optimum solution, so $|S^*| = k^*$ and $f(S^*) \geq Q$. By monotonicity, we get that $f(S^* \cup S_j) \geq Q$. Let $S^* \backslash S_j = \{w_1, ...w_d\}$. We know that the marginal gain that $w_i$ gives to a subset of $S_j$ is less than $\frac{Q}{2^j}$ for all $1 \leq i \leq d$. By submodularity, we get that $\Delta(w_i|S_j) < \frac{Q}{2^j}$ for all $1 \leq i \leq d$. Therefore,

$$f(S^* \cup S_j) - f(S_j) \leq \sum_{i=1}^{d} \Delta(w_i|S_j) < d\frac{Q}{2^j} \leq d\frac{Q}{k^* \ln 1/\epsilon} \leq \frac{Q}{\ln 1/\epsilon},$$

which gives

$$f(S^* \cup S_j) - f(S_j) \leq \frac{Q}{\ln 1/\epsilon} \rightarrow f(S_j) \geq Q\left(1 - \frac{1}{\ln 1/\epsilon}\right).$$

Hence the set $S_j$ is indeed a good candidate solution. Therefore, the algorithm does not return any solution, only if the assumption is not satisfied, i.e., $k^* \ln 1/\epsilon > M$. Otherwise, if the algorithm returns a solution $S_{j_1}$, such that $j_1 < j$, then $S_{j_1}$ must satisfy $f(S_{j_1}) \geq (1 - \frac{1}{\ln 1/\epsilon})Q$, and it contains fewer elements than $2^j$, then it also satisfies size and utility guarantees. $\square$

## B  Lower Bound

In this section, we prove an unconditional lower bound on any $p$-pass streaming algorithm that approximates the streaming submodular cover problem, as stated in Theorem 2. A common way to prove lower bounds for streaming algorithms is through communication complexity. For our purposes, the suitable communication problem to start with is the Multi-Player Pointer Jumping Problem.

In what follows, we start by defining the Multi-Player Pointer Jumping Problem and stating its known communication complexity hardness results in Section B.1. We then present in Section B.2 a reduction from the streaming submodular cover problem to the aforementioned communication problem. The proof of Theorem 2 then follows in Section B.3.

### B.1  Multi-Player Pointer Jumping Problem

In the Multi-Player Pointer Jumping Problem, we are given a rooted tree $T$ of depth $\ell \geq 1$, where the nodes in the tree are divided into $k = \ell + 1$ layers according to their distance from the root, with the convention that the root is at layer $k$, and the leaves are at layer 1. For $1 \leq i \leq k$, we refer to the set of nodes in layer $i$ as $\mathcal{V}_i$. In this problem, denoted $\text{MPJ}_{T,k}$ for a given tree $T$, we have $k$ players $P_1, \ldots, P_k$ where each player $i$ is *in charge* of the vertices in layer $i$. An input $\pi$ for $\text{MPJ}_{T,k}$ is divided as follows:

- For each $2 \leq i \leq k$, and for each $v \in \mathcal{V}_i$, player $P_i$ gets a pointer indicating one outgoing edge from $v$ to one of its children, say $u$, in layer $i - 1$, and we write in this context that $\pi(v) = u$.
- For each $v \in \mathcal{V}_1$, player $P_1$ gets a bit $b(v) := b_\pi(v) \in \{0, 1\}$.

Note that given $\pi$, the tree $T_\pi$ restricted to the edges indicated in $\pi$ contains a unique root-to-leaf path, leading to a unique leaf $v_\pi \in \mathcal{V}_1$, and hence the output of this problem given an input $\pi$ is $\text{MPJ}_{T,k}(\pi) = b(v_\pi) \in \{0, 1\}$.

This problem is interesting from a communication protocol perspective, where the $k$ players get the input $\pi$ as defined earlier, and broadcast messages in $r$ rounds. In each round, players $P_1, P_2, \ldots, P_k$ each send a message, in that order. In this setting, the message in the last round is a single bit, corresponding to their guess about $\text{MPJ}_{T,k}(\pi)$. The goal in this problem then is to figure out this bit using the minimum amount of communication per round. At each round, we think of the players as writing their messages in the order from $P_1$ to $P_k$ on a shared blackboard. Formally speaking, an $[r, C, \epsilon]$-protocol for $\text{MPJ}_{T,k}$ is defined as

- The game is played in $r$ rounds.
- The total number of bits communicated per round is at most $C$.
- The protocol's output is equal to $\text{MPJ}_{T,k}(\pi)$ with probability at least $1 - \epsilon$.

We quantify the *complexity* of such communication problems by studying their r-round randomised communication complexity $R^r(\text{MPJ}_{T,k})$ of $\text{MPJ}_{T,k}$ as follows:

$$R^r(\text{MPJ}_{T,k}) = \min\{C : \text{there exists a } [r, C, 1/3]\text{-protocol for } \text{MPJ}_{T,k}\}$$

A result in communication complexity shows that if the number of players is $k$, then the problem requires a large number of communicated bits if we restrct ourselves to $r$-rounds protocol for $r < k$. Formally, the authors of [6] prove the following:

**Theorem 5.** *Let $T$ be complete $t$-ary tree of depth $\ell \geq 1$ (and consequently $k = \ell + 1 \geq 2$ layers). Then*

$$R^{k-1}(\text{MPJ}_{T,k}) = \Omega\left(\frac{t}{k^2}\right)$$

The $\text{MPJ}_{T,k}$ problem will serve as the starting hard communication problem that we use to prove our streaming lower bounds for the submodular cover problem. In what follows, we will prove a lower bound on the required memory of any $p$-pass streaming algorithm with good approximation guarantee for the submodular cover problem. To do that, let $m$ be the number of sets that arrive in the stream, and $p$ be the number of passes that the streaming algorithm is allowed to perform. We will show that if a $p$-pass streaming algorithm can provide an approximation guarantee smaller than $m^{\frac{1}{p}}/p$ for the the set cover problem using a memory less than $\Omega(m^{\frac{1}{p}}/p^3)$, then we can solve the Multi-Player Pointer Jumping Problem on any complete $t$-ary tree with $p + 1$ player in $p$ rounds while communicating less than $\Omega(t/p^2)$ bits per round, which yields a contradiction to Theorem 5. We formalise this in what follows.

## B.2   Reduction to Submodular Cover

Let $T := T(t, k)$ be a complete $t$-ary tree with $k$ layers over the vertex set $\mathcal{V}$, where each layer contains vertices $\mathcal{V}_i \subseteq \mathcal{V}$ for $1 \leq i \leq k$. Note that

$$|\mathcal{V}_i| = t^{k-i} \qquad\qquad \forall i \in \{1, 2, \ldots, k\}$$

$$|\mathcal{V}| = \sum_{i=1}^{k} |\mathcal{V}_i| = \sum_{i=1}^{k} t^{k-i} = \frac{t^k - 1}{t - 1}$$

Without loss of generality, assume the children of every node $v \in \bigcup_{i=2}^{k} \mathcal{V}_i$ are ordered, i.e., for every $2 \leq i \leq k$, for every $v \in \mathcal{V}_i$, and for every $1 \leq j \leq t$, $c_j(v) \in \mathcal{V}_{i-1}$ denotes the $j^{th}$ child of $v$ in layer $i - 1$.

Given a complete $t$-ary tree $T$ with $k$ layers, we construct a structure $\mathcal{T}_{t,k,\ell}$ for $\ell \geq t$, such that $\mathcal{T}$ has the same tree structure as $T$, but we additionally have sets associated with each $v \in \mathcal{V}$. To do that, let $\mathcal{X}$ be a universe of elements such that $|\mathcal{X}| = t^{k-1}\ell$. We partition $\mathcal{X}$ into $t^{k-1}$ equal sized (i.e., each of size $\ell$) disjoint sets, and assign each such set $\tilde{S}$ to one of the $t^{k-1}$ leafs of $T$ (i.e., the vertices in $\mathcal{V}_1$). For every vertex $v \in \mathcal{V}_1$, we denote its corresponding set by $\tilde{S}_v$. This process defines the sets assigned to leaf vertices. We now recursively define the sets associated with the remaining vertices as follows:

- For every $i = 2, \ldots, k$, and for every $v \in \mathcal{V}_i$, the set $\tilde{S}_v$ associated with the vertex $v$ is the union of the sets corresponding to its children, i.e.,

$$\tilde{S}_v = \bigcup_{j=1}^{t} \tilde{S}_{c_j(v)}$$

Note that with this construction, if we denote by $r$ the root of the tree (i.e., the unique vertex in $\mathcal{V}_k$), then $\tilde{S}_r = \mathcal{X}$. In fact, one can easily verify that for any $i \in \{1, 2, \ldots, k\}$, and any $v \in \mathcal{V}_i$, our construction satisfies that

$$|\tilde{S}_v| = \ell t^{i-1} \tag{3}$$

Before we proceed, we record the following easy observation.

**Observation 1.** *For any 2 vertices $v, u \in \mathcal{V}$ such that $v$ is neither the ancestor nor the descendent of $v$, we have that $\tilde{S}_v \cap \tilde{S}_u = \emptyset$.*

Now given an input $\pi$ for $\mathrm{MPJ}_{T,k}$ over $T := T(t, k)$, we construct an instance $\mathcal{I}(\pi)$ of $\mathrm{SMC}_{t,k,\ell}$ using $\mathcal{T}^{t,k,\ell}$ as follows, where each player out of the $k$ players $P_1, P_2, \ldots, P_k$ gets a collection of sets. Namely,

- Player $P_k$ gets the following sets; let $r$ be the root of the tree, and let $u = \pi(r)$ be the chosen child of $v$ according to $\pi$, then $P_k$ gets the set $S_r = \tilde{S}_r \backslash \tilde{S}_u$, and a singleton set for each element $e \in \tilde{S}_u$. It follows from (3) that $P_k$ gets $\ell t^{k-2} + 1$ sets.

- For each $2 \leq i \leq k-1$, player $P_i$ gets the following sets: For each $v \in \mathcal{V}_i$, let $u = \pi(v)$ be the chosen child of $v$ according to $\pi$. Then Player $P_i$ gets for each such vertex $v$ the set $S_v = \tilde{S}_v \backslash \tilde{S}_u$. Hence, for $2 \leq i \leq k-1$, player $P_i$ gets in total $t^{k-i}$ sets.

- Player $P_1$ gets the following sets: For each $v \in \mathcal{V}_1$, player $P_1$ gets the set $S_v = \tilde{S}_v$ if $b(v) = 1$, and nothing from this vertex if $b(v) = 0$. Hence $P_1$ gets at most $t^{k-1}$ sets.

Let $\mathcal{S}_i$ be the family of sets that player $P_i$ gets, and $\mathcal{S}$ be the family of all the sets in our instance, then $m := |\mathcal{S}|$ is at most

$$m \leq \underbrace{t^{k-1}}_{P_1} + \underbrace{\sum_{i=2}^{k-1} t^{k-i}}_{P_2, \ldots, P_{k-1}} + \underbrace{\ell t^{k-2} + 1}_{P_k} = \frac{t^k - 1 + \ell t^{k-2}(t-1)}{t-1} \text{ sets.}$$

The submodular function $f$ that we consider for $\mathrm{SMC}_{t,k,\ell}$ is the cover function defined as follows:

$$f(S) = |S| \qquad\qquad \forall S \in \mathcal{S}$$
$$f(\tilde{\mathcal{S}}) = |\bigcup_{S \in \tilde{\mathcal{S}}} S| \qquad\qquad \forall \tilde{\mathcal{S}} \subseteq \mathcal{S}$$

Then the goal in this submodular cover problem is to find the minimum cardinality family of sets $\mathcal{S}^* \subset \mathcal{S}$ such that

$$f(\mathcal{S}^*) \geq \ell t^{k-1} := Q$$

Note that in our construction, we can always achieve the desired value $Q$, no matter whether $\mathrm{MPJ}_{T,k}(\pi)$ is 0 or 1, as the sets assigned to player $P_k$ are enough to cover $\mathcal{X}$. In other words, one can easily check that $f(\mathcal{S}_k) = \ell t^{k-1} = Q$.

In order to see the intuition behind this construction, recall that in the communication problem $\mathrm{MPJ}_{T,k}$, we were interested in the value of $\mathrm{MPJ}_{T,k}(\pi) = b(v_\pi)$ given an input $\pi$. We will show in the following claim how to relate the value of $b(v_\pi)$ to the size of the submodular cover in the instance $\mathcal{I}(\pi)$ of $\mathrm{SMC}_{t,k,\ell}$ that we have constructed.

**Claim 1.** *Let $\mathrm{MPJ}_{T,k}$ be an instance of the Multi-Player Pointer Jumping Problem, and let $\mathcal{I}(\pi)$ be the resulting submodular cover instance of $\mathrm{SMC}_{t,k,\ell}$. Then the following holds:*

1. *If $b(v_\pi) = 1$, then there exists a family of sets $\mathcal{S}$ such that $|\mathcal{S}| = k$, and $f(\mathcal{S}) = Q$.*

2. *If $b(v_\pi) = 0$, then any family of sets $\mathcal{S}$ of size less than $\ell$ will have $f(\mathcal{S}) < Q$.*

*Proof.* Let $v_k, v_{k-1}, \ldots, v_1$ be the unique root-to-leaf path resulting from $\pi$.

We start with the first case, i.e., $b(v_\pi) = 1$. In this case, consider the sets $S_{v_k}, S_{v_{k-1}}, \ldots, S_{v_1}$. It follows from the definition of the sets according to $\pi$ that

$$
\begin{aligned}
S_{v_k} &= \tilde{S}_{v_k} \backslash \tilde{S}_{v_{k-1}} = \mathcal{X} \backslash \tilde{S}_{v_{k-1}} \\
S_{v_{k-1}} &= \tilde{S}_{v_{k-1}} \backslash \tilde{S}_{v_{k-2}} \\
\vdots \;\; &= \;\; \vdots \\
S_{v_2} &= \tilde{S}_{v_2} \backslash \tilde{S}_{v_1}
\end{aligned}
$$

where $\tilde{S}_{v_1} = S_{v_1}$ since $b(v_1) = b(v_\pi) = 1$ in $\pi$. It then follows that

$$
\bigcup_{i=1}^{k} S_{v_i} = \mathcal{X}
$$

and hence for $\mathcal{S} = \left\{ S_{v_k}, S_{v_{k-1}}, \ldots, S_{v_1} \right\}$, we get $f(\mathcal{S}) = Q$.

Now for the second case, also consider $v_k, v_{k-1}, \ldots, v_1$ the unique root-to-leaf path resulting from $\pi$. In this case, since $b(v_\pi) = b(v_1) = 0$, $S_{v_1} = \emptyset$. Note however that $\tilde{S}_{v_1}$ contained $\ell$ elements. We claim that for any non-singleton set $S \in \mathcal{S}$, $S \cap \tilde{S}_{v_1} = \emptyset$. To see this, observe that for any vertex $v$, $S_v \subseteq \tilde{S}_v$, and hence we never add to $S_v$ an element that was not already in $\tilde{S}_v$. Thus, Observation 1 yields that for any $S_u \in \mathcal{S}$ such that $u$ is not an ancestor of $v_1$, $S_u \cap \tilde{S}_{v_1} = \emptyset$. It remains to show that $S_{v_i} \cap \tilde{S}_{v_1} = \emptyset$ for all $2 \leq i \leq k$, i.e., for the set associated with the vertices on the unique root-to-leaf path. But this easily follows from the *child-exclusion nature* of our construction , that yields that $\tilde{S}_{v_1}$ does is disjoint from any set on the path leading from the root to $v_1$.

This says that in order to cover the elements of $\tilde{S}_{v_1}$, we need to include all the required singletons (recall that these singletons must be in $\mathcal{S}_k$), and hence since $|\tilde{S}_{v_1}| = \ell$, we need to include at least $\ell$ singleton sets, which yields the second part of the claim. $\qquad\square$

In our argument for the second case, we assumed that we can cover $\mathcal{X} \backslash S_{v_1}$ *for free*. One can refine the analysis to show that in this case, any family of sets $\tilde{\mathcal{S}}$ that achieves $f(\tilde{\mathcal{S}}) \geq Q$ must have a size of at least $\ell + k - 1$. For our purposes, we think of $\ell >> k$, and hence a gap between $k$ and $\ell$ is enough for the main result of this section.

### B.3 Lower bound for the Submodular Cover Problem

We are now ready to prove the main theorem of this section.

**Theorem 6.** *For any number of passes $p$ and any stream size $m$, a $p$-pass streaming algorithm that, with probability at least 2/3, approximates the submodular cover problem to a factor smaller than $\frac{m^{\frac{1}{p}}}{p+1}$ must use at least $\Omega\left( \frac{m^{\frac{1}{p}}}{p(p+1)^2} \right)$.*

*Proof.* Let $\mathrm{MPJ}_{T,p+1}$ be an instance of the Multi-Player Pointer Jumping Problem over a complete $t$-ary tree $T$ with $p+1$ levels, and let $\pi$ be the input to the problem spread amongst the $p+1$ players $P_1, P_2, \ldots, P_{p+1}$ as discussed in Section B.1. We now construct the instance $\mathcal{I}(\pi)$ of $\mathrm{SMC}_{t,p+1,\ell}$ for some integer $\ell \geq t$ over $m$ sets as described in Section B.2. Recall that

$$
m \leq \frac{t^k - 1 + \ell t^{k-2}(t-1)}{t-1}
$$

and $k = p + 1$ in our case.

It follows from Claim 1 that distinguishing between the case when $b(v_\pi) = 1$ and $b(b_\pi) = 0$ is equivalent to distinguishing whether the minimal set cover $\mathcal{S}^*$ of $\text{SMC}_{t,p+1,\ell}$ is of size at most $p+1$, or at least $\ell$.

Consider a $p$-pass streaming algorithm ALG that approximates $\text{SMC}_{t,p+1,\ell}$ using memory $M$, to a factor smaller than $\frac{\ell}{p+1}$, where the stream consist of the sets of $P_1$ followed by those of $P_2$ and so on. We will use ALG to design the following an $[p, (p+1)M, 1/3]$-protocol PRTCL for $\text{MPJ}_{T,k}$ as follows:

- In each round $i = 1, \ldots, p$, we emulate the $i^{th}$ pass of ALG on the stream; When ALG processes the last set corresponding to $P_1$ in the stream, the content of the memory is broadcasted to all the players. Then we do the same after ALG finishes $P_2$'s chunk on the stream, and so on up to $P_{p+1}$, in that order.

Since ALG approximates the size of $\mathcal{S}^*$ for $\mathcal{I}(\pi)$ to a factor smaller than $\frac{\ell}{p+1}$ with probability at least $2/3$, then PRTCL outputs $\text{MPJ}_{T,k}(\pi)$ with probability at least $2/3$. Recall that the game $\text{MPJ}_{T,p+1}$ is played among $p+1$ players, and we know from Theorem 5 that $R^p(\text{MPJ}_{T,p+1}) = \Omega\left(\frac{t}{(p+1)^2}\right)$, hence $M$ must be at least $M = \Omega\left(\frac{t}{p(p+1)^2}\right)$.

It remains to relate the approximation ratio and the required memory size, to the size of the stream $m$. Recall that:

$$m \leq \frac{t^k - 1 + \ell t^{k-2}(t-1)}{t-1}$$

For $\ell = t$, we get that $m \leq \frac{2t^k - t^{k-1}}{t-1} \leq 2t^{k-1} = 2t^p$, and equivalently $\ell = t \approx m^{1/p}$. Thus we get that any $p$-pass streaming algorithm that, with probability at least $2/3$, approximates the submodular cover problem to a factor smaller than $\frac{m^{\frac{1}{p}}}{p+1}$ must use at least $\Omega\left(\frac{m^{\frac{1}{p}}}{p(p+1)^2}\right)$ memory. $\qquad\square$