[Reviews · NeurIPS 2016]

Reviewer 1

Summary

This paper initiates the study of the submodular coverage problem in the streaming setting. In the problem, there is a universe of elements U. The elements of the universe arrive one at a time and the algorithm has limited memory to store elements. The is a positive, monotone, submodular function f. The goal is to construct a set S of size at most k (where k is a parameter) such that f(S) has at least some value Q. The problem is a generalization of the streaming set cover problem. The paper shows that any algorithm cannot have bounded performance for the problem without unrealistic assumptions (which I strongly agree with) on the memory size. Due to this, they consider relaxing the problem. In the relaxed version of the problem, one only needs to find a set that has value at least (1-\eps)Q and has size \detla k where \eps and \delta are some constants. The theoretical results are not too challenging to show and follow from ideas given in previous work on the set cover problem. Similar results to theirs are known for set cover. The authors show that on a variety of data sets, their algorithms perform well in regards to the objective value. I think this is a reasonable paper that can be accepted.

Qualitative Assessment

The paper introduces a reasonable problem and is able to construct algorithms with good guarantees for the problem.

Confidence in this Review

2-Confident (read it all; understood it all reasonably well)


Reviewer 2

Summary

This paper studies the streaming version of set cover. In the model proposed here, the input consists of sets that cover a universe that arrive online in an arbitrary order, and an algorithm must find an approximately optimal solution using a single pass on the input while using sublinear memory. The main result shows that if one is willing to compromise for a 1-\epsilon coverage of the universe, there is an algorithm that achieves (log n/\epsilon) approximation of the offline optimal solution. Thus, the algorithm is 2/\epsilon factor of optimal, as the offline set cover problem cannot be approximated within a factor strictly better than log n. In addition, the authors justify the relaxed version of set cover with a nice lower bound. They show that obtaining a non-trivial approximation guarantee for the online set cover problem (i.e. without the 1-\epsilon relaxation) requires linear memory.

Qualitative Assessment

The problem studied is the min set cover where there is a family of sets that cover some universe and the goal is to find a family of sets of minimal size that covers at least Q elements in the universe. It is well known that this problem is NP hard and that no polynomial time algorithm can obtain an approximation guarantee better than log n, under reasonable computational complexity assumptions. In the streaming setting, defined in this paper for this problem for the first time, the goal is to solve the set cover problem using a single pass on the data using sublinear memory when the assumption is that sets arrive in an adversarial order. Motivated by impossibility results for this problem, shown in this paper, the authors define the Streaming Bicriteria Submodular Cover (SBSC) problem where the goal is to obtain (1-\epsilon,delta) bi-criteria approximations, i.e. cover a (1-\epsilon) fraction of the universe using \delta factor of the minimum sets of the optimal solution. The main result is a streaming algorithm in the model defined which covers a (1-\epsilon) fraction of universe using 2/\epsilon sets of the optimal solution. The idea of the analysis is a refinement of the classic analysis of the greedy algorithm for set cover by Wolsey. In particular, as the authors point out, if the greedy algorithm is stopped after covering 1-\epsilon of the universe, this implies that at most k*ln(1/\epsilon) sets are picked which gives a tight (1-\epsilon,ln(1/\epsilon)) bi criteria offlne apx algorithm. The authors essentially take this idea and turn it into a streaming algorithm by maintaining log n different solutions, each with a different threshold (in multiples of 2). An element is then added to all the solutions in which its coverage passed the threshold of the solution. At a high level, the idea of the algorithm is very similar to previous streaming algorithms for the dual max-k-cover problem. The analysis however, introduces new ideas as discussed above. In terms of applications, the authors give two nice applications of active set selection and graph set cover. Both are problems studied in the past where it is quite reasonable to use a streaming algorithm. The experiments are quite comprehensive, and are preformed on very large data sets (graphs with nearly a billion nodes and ~50 billon edges). The algorithm was run against strong benchmarks, and seems to be competitive with the offline greedy algorithm which is optimal for this problem. Overall, this is a nice paper. The paper introduces a new problem which is quite natural and does a great job in modeling given that the strict version of the problem does not yield non-trivial solutions. The algorithms are similar to previous ones for related problems, but the ideas guiding the analysis are all new and interesting. The applications are interesting and the experiments are solid. In addition, the paper is well written and easy to follow. Minor comments: - In line 47, shouldn't the main result be 2/\epsilon (rather than 1/\epsilon) of the minimal number of sets?

Confidence in this Review

3-Expert (read the paper in detail, know the area, quite certain of my opinion)


Reviewer 3

Summary

This paper deals with submodular cover problem. One is given a function f defined on subsets of a set V to reals. It is assumed that f is monotone, submodular and normalized (f(empty_set)=0). Given a target Q, the problem then is find a set S \subset V of minimum cardinality such that f(S)>= Q. This is an important problem with a significant recent activity. The main contribution of this paper is that it gives a (bi-criteria ) approximation algorithm for this problem in the streaming setting. They first show (they state a more general theorem) that why a single pass algorithm cannot give any non-trivial result in the usual sense of approximation. They then give an algorithm for a relaxed notion of approximation which they refer to as bi-criteria approximation. They note that this approximation notion has been studied in the setting of semi-streaming setting for the set cover problem.

Qualitative Assessment

This is an important problem and with the presence large data sets, studying this in a streaming setting seems very useful. The paper is well written, and it is clear from Theorem 2 that usual notion of approximation doesn't have any single-pass algorithm giving non-trivial guarantees. They do compare the guarantees with the greedy offline algorithm. In certain regime they argue that the size of the set output by the single-pass streaming algorithm given in the paper is at most twice that output by the offline version, with a similar approximation guarantee. It is not clear however what is the lower bound and how close to optimal the algorithm given in this paper is in general.

Confidence in this Review

1-Less confident (might not have understood significant parts)


Reviewer 4

Summary

In this paper, the authors study the streaming submodular cover problem: given a monotone submodular function f and a target utility Q, the goal is to find a set S of minimal size such that f(S) is at least Q. The setting is the one of streaming algorithms where the goal is to perform a small (constant) number of passes over the data stream, and use sublinear memory (i.e the algorithm cannot simply store the entire stream of data). The first result is an impossibility result: any non-trivial approximation guarantee (better than m/2 where m is the size of the stream) requires linear memory. This first result motivates the formulation of a bi-criteria formulation of the streaming submodular cover problem: find a set of size at most delta times the optimal set which gets a fraction (1-epsilon) of the optimal solution. The authors provide an algorithm which provably obtains this, with delta = 1/epsilon.

Qualitative Assessment

The paper is well-written and the results are correct as far as I checked (I didn't read the proofs in the appendix but checked the ones in the main body of the paper). The experiments are interesting and illustrates well both the scalability of the approach and the trade-off introduced by the parameter alpha in the algorithm. My main concern is a conceptual interrogation regarding the computational model: I think it is hard to formulate a sound computation model for streaming optimization of arbitrary submodular functions. Indeed, since these functions can have exponential size representations in the general case, the authors assume access to a value oracle (as is standard in the offline setting). It is not clear to me what such oracles imply in the streaming setting: for example, the algorithm could cheat and use value queries to avoid having to store items. The exact interplay between the value queries and the memory limitation introduced by the streaming model should be properly discussed. For specific examples (like submodular set cover which was previously studied in the streaming setting, or the ones discussed in the experiments), this problem does not exist since the function has an explicit representation which can be computed as the strem of elements is received, but I think the general case requires a more in-depth discussion of the computational model. Finally, while the lower-bound is novel and technically interesting, the algorithm and its analysis seem to be an incremental variation on a well-known technique: it has already been observed in other settings (mapreduce submodular optimization for example) that the following simple threshold-based algorithm gives constant approximation: for decreasing values of the threshold (divide by a constant at each iteration) include in the solutions all elements whose marginal contribution is above the threshold).

Confidence in this Review

2-Confident (read it all; understood it all reasonably well)


Reviewer 5

Summary

The paper proposes an algorithm for approximately solving the streaming submodular cover problem (in a bicriterion sense). Furthermore, the trade-off between memory and the approximation quality achieved by the proposed algorithm is analyzed. The effectiveness of the algorithm is demonstrated in large scale experiments.

Qualitative Assessment

I enjoyed reading this mainly well written paper and think that the considered problem as well as the obtained results are interesting. I only have a few questions: * Would there be any value in replacing items picked in an early stage of the algorithm by more valueable items that are observed later? This would not increase the (bound on the) runtime of the algorithm and the required memory would be the same up to a constant. * Is there an explanation for the "jumpy" trends in Figure 1b? Are the results averaged over multiple orderings of the elements? Minor comments: * Line 96: f: V -> ... f:2^V -> * Line 137: add reference to algorithm 2 * Line 140: Not that in this * Line 145: where k^* is the size of an optimal... * Line 175: remove "a" * Line 186: Shouldn't the ln 1 / \epsilon be in the denominator? * Line 187: at most 2k^*\log 1 / \epsilon? * Line 190: marginal gain * Line 193: shown * Line 231: with vertex set V * Line 261: Figure 6.1 -> Figure 1 * Line 262: the offline greedy * Line 269: \epsilon -> \alpha * Line 293: the dominating set problem * Line 298: reference to figure 6.2

Confidence in this Review

2-Confident (read it all; understood it all reasonably well)